# Pembrolizumab as Consolidation Strategy in Patients with Multiple Myeloma: Results of the GEM-Pembresid Clinical Trial

**DOI:** 10.3390/cancers12123615

**Published:** 2020-12-03

**Authors:** Noemí Puig, Luis A. Corchete-Sánchez, José J. Pérez-Morán, Julio Dávila, Teresa Paíno, Javier de la Rubia, Albert Oriol, Jesús Martín-Sánchez, Felipe de Arriba, Joan Bladé, María-Jesús Blanchard, Verónica González-Calle, Ramón García-Sanz, Bruno Paiva, Juan-José Lahuerta, Jesús F. San-Miguel, María-Victoria Mateos, Enrique M. Ocio

**Affiliations:** 1Cancer Research Center (CiC-IBMCC, CSIC/USAL), Hematology Department, Center for Biomedical Research in Network of Cancer (CIBERONC), Institute of Biomedical Research of Salamanca (IBSAL), University Hospital of Salamanca, 37008 Salamanca, Spain; npuig@saludcastillayleon.es (N.P.); lacorsan@usal.es (L.A.C.-S.); jpmoran@saludcastillayleon.es (J.J.P.-M.); tpaino@usal.es (T.P.); vgcalle@saludcastillayleon.es (V.G.-C.); rgarcias@usal.es (R.G.-S.); mvmateos@usal.es (M.-V.M.); 2Complejo Asistencial de Ávila, Hematology Department, 05071 Ávila, Spain; jdavila@saludcastillayleon.es; 3Hematology Service, University Hospital Peset and Internal Medicine, School of Medicine and Dentistry, Catholic University of Valencia, 46017 Valencia, Spain; delarubia_jav@gva.es; 4Hematology Department, Institut Català d’Oncologia and Institut Josep Carreras, Hospital Germans Trials i Pujol, 08916 Badalona, Barcelona, Spain; aoriol@iconcologia.net; 5Hematology Department, Hospital Universitario Virgen del Rocío, 41013 Sevilla, Spain; jesus.martin.sspa@juntadeandalucia.es; 6Servicio de Hematología y Oncología Médica, Hospital Universitario Morales Meseguer, IMIB-Arrixaca, Universidad de Murcia, 30008 Murcia, Spain; Farriba@um.es; 7Hematology Department, Hospital Clinic, Institut d’Investigacions Biomèdiques August Pi I Sunyer, 08036 Barcelona, Spain; jblade@clinic.cat; 8Hematology Department, Hospital Ramón y Cajal, 28034 Madrid, Spain; mariajesus.blanchard@salud.madrid.org; 9Centro de Investigación Médica Aplicada, Instituto de Investigación Sanitaria de Navarra, Clínica Universidad de Navarra, CIBERONC, 31008 Pamplona, Spain; bpaiva@unav.es (B.P.); sanmiguel@unav.es (J.F.S.-M.); 10Centro de Investigación Biomédica en Red Cáncer (CIBERONC), Instituto de Investigación, Hospital Universitario 12 de Octubre, 28041 Madrid, Spain; jjLAHUERTA@telefonica.net; 11Hospital Universitario Marqués de Valdecilla (IDIVAL), Universidad de Cantabria, 39008 Santander, Spain

**Keywords:** pembrolizumab, immunotherapy, myeloma, consolidation

## Abstract

**Simple Summary:**

Multiple myeloma patients with persistent disease after treatment show increased expression of PDL1 in tumor plasma cells and of PD1 in T lymphocytes. This suggests a role of the PD1/PDL1 axis in treatment failure that could potentially be reverted with pembrolizumab, an anti-PD1 monoclonal antibody. The GEM-Pembresid trial enrolled 20 patients with multiple myeloma achieving a suboptimal response to the previous treatment that received intravenous pembrolizumab every 3 weeks with the objective of eradicating the residual disease. Pembrolizumab was acceptably well tolerated in the 17 patients evaluable for safety, but no improvement in the baseline responses was documented. Although no determinants of response could be identified, we detected a lower expression of PD1/PDL1 in a subgroup of patients progressing in the first 4 months after enrollment; furthermore, a reduction in the percentage of NK cells induced by pembrolizumab was observed.

**Abstract:**

PD1 expression in CD4^+^ and CD8^+^ T cells is increased after treatment in multiple myeloma patients with persistent disease. The GEM-Pembresid trial analyzed the efficacy and safety of pembrolizumab as consolidation in patients achieving at least very good partial response but with persistent measurable disease after first- or second-line treatment. Moreover, the characteristics of the immune system were investigated to identify potential biomarkers of response to pembrolizumab. One out of the 17 evaluable patients showed a decrease in the amount of M-protein, although a potential late effect of high-dose melphalan could not be ruled out. Fourteen adverse events were considered related to pembrolizumab, two of which (G3 diarrhea and G2 pneumonitis) prompted treatment discontinuation and all resolving without sequelae. Interestingly, pembrolizumab induced a decrease in the percentage of NK cells at cycle 3, due to the reduction of the circulating and adaptive subsets (0.615 vs. 0.43, *p* = 0.007; 1.12 vs. 0.86, *p* = 0.02). In the early progressors, a significantly lower expression of PD1 in CD8^+^ effector memory T cells (MFI 1327 vs. 926, *p* = 0.03) was observed. In conclusion, pembrolizumab used as consolidation monotherapy shows an acceptable toxicity profile but did not improve responses in this MM patient population. The trial was registered at clinicaltrials.gov with identifier NCT02636010 and with EUDRACT number 2015-003359-23.

## 1. Introduction

Checkpoint inhibitors, such as anti-PD1/PDL1 monoclonal antibodies (MoAbs), have become a promising strategy in oncology, with the approval of some of them for the treatment of different tumors [1]. In multiple myeloma (MM), the activity of nivolumab in monotherapy was initially tested in the context of a phase Ib study that included 27 relapsed/refractory MM patients (RRMM); interestingly, one patient responded, achieving a complete response (CR) [1]. Later on, the combination of another anti-PD1 MoAb, pembrolizumab, with either pomalidomide [2] or lenalidomide [3], was tested in phase II trials with promising results. However, phase III trials of pembrolizumab in combination with the same agents (KEYNOTE-183 (NCT02576977) and KEYNOTE-185 (NCT02579863)) [3,4] were prematurely stopped due to a survival disbalance disfavoring those patients receiving the MoAbs. These negative results were attributed to a potential antagonist effect of these combinations. Thus, alternative strategies or novel partners for combination are currently being explored with these and other checkpoint inhibitors.

In this regard, an important field of investigation is the search for biomarkers that could predict sensitivity or resistance to these agents. Although there are still many uncertainties, several of them such as mutational load [5,6] or microsatellite instability [7,8] have been advocated. In addition, others evaluating the immune status of the patient have been proposed, as is the case for PD1 or PDL1 expression in immune or tumor cells, the presence of tumor infiltrating lymphocytes, the immunoscore, or the cancer immunogram [9,10,11]. These potential biomarkers have mostly been investigated in the field of solid tumors [12], where a sample is sometimes more difficult to obtain particularly for sequential studies. By contrast, in MM, the availability of bone marrow (BM) samples with cells in suspension facilitates the investigation of tumor cells, as well as their immune environment.

In addition to the common expression of PDL1 on most MM plasma cells (PCs) [13], our group recently showed that MM patients with persistent minimal residual disease (MRD) after treatment showed an upregulation of PDL1 in residual PCs, as well as a significant increase in PD1 expression in CD4 and CD8 T cells, suggesting that the activation of this pathway could operate as a relevant immune checkpoint in the tumor microenvironment and could be responsible for the failure to eradicate the residual MM cells [14]. On the basis of these data, we designed the present trial in order to investigate the potential role of pembrolizumab monotherapy in eradicating residual pathological PCs. Patients with very good partial response (VGPR) or better with measurable residual disease were selected to receive pembrolizumab as consolidation. Efficacy data were correlated with the distribution of different immune cell populations at baseline and their respective PD1/PDL1 expression by flow cytometry and with specific changes induced by the treatment.

## 2. Methods

### 2.1. Study Design and Patient Population

GEM-Pembresid is a national, multicenter, open-label single-arm, noncomparative study aimed at determining the efficacy of, safety of, and changes in selected immune markers induced by pembrolizumab monotherapy administered as consolidation therapy in MM patients who have achieved a response with a previous treatment but who have residual disease.

For this purpose, 20 MM patients, who had received any treatment of limited duration either at diagnosis or at first relapse, and who achieved a good response (at least VGPR) but with persistent residual disease (that is, patients in VGPR, CR, or stringent CR (sCR) with MRD positive) were treated with pembrolizumab monotherapy administered intravenously at a dose of 200 mg every 3 weeks for 1 year, with a potential expansion of 1 additional year of treatment in cases of clinical benefit and patient agreement. Efficacy, safety, and immune biomarker parameters were evaluated to understand the role of pembrolizumab in this setting. A confirmation of the stability of the residual disease in the 3 months before the initiation of treatment (amount of M-protein and/or MRD level) was required in all patients. The Institutional Review Board/Independent Ethics Committee at each participating center approved the study. The code assigned to the evaluation of this project was 15/1203. All patients provided written informed consent before screening. The trial was registered at clinicaltrials.gov with identifier NCT02636010 and with EUDRACT number 2015-003359-23.

### 2.2. Endpoints and Study Assessments

Primary endpoint was the upgrade of the response obtained with the previous therapy (VGPR to ≥ CR; CR to ≥ sCR; sCR with MRD positive to sCR with MRD negative). Secondary endpoints were clinical and laboratory toxicities and rate of discontinuation of pembrolizumab monotherapy, rate of achievement of flow MRD-negative sCR, and pharmacodynamic immune biomarker parameters (see below). Assessment of efficacy was done using the International Myeloma Working Group (imwg) criteria [15] Maximal response before entering the trial was recorded and reevaluated prior to the administration of each dose and in the end of treatment visit. In those patients in MRD-positive CR (either at the initiation of the study or in the course of the trial), bone marrow (BM) assessments and analysis of the residual disease by flow cytometry were performed prior to every two administrations of pembrolizumab (that is, prior to doses 2, 4, 6, etc.) until the potential achievement of MRD-negative CR. To be evaluable for efficacy, patients must have had baseline evaluation of disease status, at least one complete dose of treatment, and at least one follow-up evaluation of the disease. MRD assessment was performed by multiparameter flow cytometry, following the Euroflow guidelines, for a highly sensitive (2 × 10^−6^) and standardized MRD detection in MM [16]. Safety was assessed evaluating clinical and laboratory findings and classified according to the National Cancer Institute Common Terminology Criteria for Adverse Events v. 4.03 (NCI CTCAE). Patients were evaluable for safety if they had received at least one total or partial infusion of the study drug. N.P., L.C., J.J.P., T.P., and E.M.O. analyzed the data, and all authors had access to primary clinical trial data.

### 2.3. Evaluation of the Immune System

Samples were obtained at screening (peripheral blood (PB) and BM), at C3D1 (third cycle, first day; only PB), at the time of CR (PB and BM), and at the end of treatment. To identify biological markers of response or resistance to pembrolizumab, the patient’s immune system (B and T lymphocytes, NK cells, dendritic cells, monocytes, and their respective subpopulations) was characterized in BM and PB by multiparameter flow cytometry. Furthermore, the PD1/PDL1 expression in these populations, as well as PDL1 expression in residual pathological PCs, was also analyzed. Lastly, changes induced by pembrolizumab in PB samples at C3D1 were evaluated. For these studies, the following combinations of MoAbs were used:-CD138/ CD27/ CD38/ CD56/ CD45/ CD19/ PDL1/ CD81 and HLADR/ CD45/ CD16/ CD64/ CD3/ CD123/ PDL1/ CD14 to analyze PDL1 expression in residual pathological PCs and potential PDL1-expressing dendritic cells before and after treatment, respectively.-CD45RA/ CD127/ CD8/ TCRγδ/ CD25/ CD197/ CD4/ PD1, to quantify and characterize T-cell subsets and their corresponding maturation stages, as well as PD1 expression in each of them.-CD57/ CCR5/ CD314/ CD85j/ CD62L/ CD3/ CD16/ CD56 to quantify and characterize NK cell subsets (circulating, native, adaptive and canonical) and to analyze their respective pattern of expression of activation/inhibition surface markers.

### 2.4. Statistical Analysis

Descriptive statistical analyses were obtained for demographic and baseline variables. Survival analyses were conducted by the Kaplan–Meier method and the log-rank test to evaluate the statistical value of the comparisons. Statistical association of the immune biomarkers with response categories codified as two groups was performed using the two-sided exact Wilcoxon–Mann–Whitney test and the two-sided exact Wilcoxon–Pratt signed rank test from the coin package (v. 1.3-1) in R (v.3.5.1) for unpaired and paired data, respectively. The nonparametric Kruskal–Wallis test followed by the Dunn’s post hoc test was carried out using the Dunn test (v. 1.3.5) R package when three or more response categories were compared.

### 2.5. Data Sharing Statement

For original data, please contact ocioem@unican.es. Deidentified individual participant data are available indefinitely at ocioem@unican.es.

## 3. Results

### 3.1. Patient Population and Characteristics

Between September 2016 and June 2017, 35 patients signed the informed consent form and were screened for the trial. Fifteen of them were initially identified as screening failures due to either absence of pathological PCs by flow cytometry in the screening BM (*n* = 10) or having a nonstable response (*n* = 5). Therefore, 20 patients started treatment and were evaluable for toxicity. Later on, 17 of them were considered evaluable for efficacy; one was finally considered to be MRD-negative at screening (discontinued in CR after one cycle) and two patients were non-evaluable because they were found to have an already progressively increasing disease when enrolled in the trial.

The features of the 20 included and the 17 efficacy-evaluable patients are summarized in Table 1. Focusing on the 17 efficacy-evaluable cases, median age was 63 (44–78), 24% of them were Bence-Jones, and approximately one-third of patients each were International Staging System (ISS) I, II, or III at diagnosis. Most of them were enrolled in the study during the first line of treatment but four (23%) were enrolled after the second line. Fourteen (80%) had received an autologous stem-cell transplantation (ASCT) as the immediately previous line before being included in the trial. In the 14 patients that had received prior ASCT, the median treatment-free interval before starting pembrolizumab was 114.5 (range, 92–139) days. Regarding the disease status at screening, eight (47%) had achieved VGPR, four (24%) had achieved CR, and five (29%) had achieved sCR with MRD positive.

### 3.2. Efficacy

At the data cutoff (15 July 2019), patients had received a median of 17 cycles of pembrolizumab (range, 1–34) (Figure 1). Regarding the primary endpoint of the study, only one patient improved the baseline response after treatment. He entered the trial having achieved VGPR with a residual paraprotein of 0.2 g/dL following VTD (bortezomib, thalidomide, and dexamethasone) plus ASCT administered as the first line of therapy. Pembrolizumab was started on day +122 after ASCT, and a progressive reduction in the amount of M-protein was observed with its administration, particularly after the initial cycles. The monoclonal component disappeared in the serum protein electrophoresis, but immunofixation persisted positive throughout and MRD remained constantly detectable (and, therefore, never achieved CR and remained classified as VGPR/near CR). This response was maintained after having finished the 2 years of treatment planned, 30 months after entering the trial. Although this clinical behavior could have been induced by pembrolizumab, the pattern of response (mainly happening after the first cycles) and its limited quality (never reaching CR) does not allow us to rule out a potential late therapeutic effect of high-dose melphalan. The remaining 16 efficacy-evaluable patients did not show any response to pembrolizumab consolidation. The median progression-free survival (PFS) was only 14 months (95% confidence interval: 0.3–27.7). All patients have already discontinued: 11 due to progression, two after completing the 2 years of treatment, two because of toxicity (grade 3 diarrhea and grade 2 pneumonitis), and two based on investigator decision (one according to the protocol after the first year of treatment, and the other one after the first cycle). Intriguingly, five of the 11 progressions occurred during the first six cycles of treatment, that is, in the first 4 months after treatment initiation. There were no differential characteristics regarding the baseline features or the previous treatments between these early progressors and the remaining group (data not shown).

### 3.3. Safety

Regarding safety, 14 adverse events (AEs) were considered to be related to pembrolizumab treatment by the investigators. Three patients developed diarrhea, two of them grade 1 and the third one grade 3, the latter leading to the discontinuation of the investigational product. One patient had grade 2 pneumonitis at cycle 31 that was considered possibly related to pembrolizumab and prompted patient discontinuation. Treatment with broad-spectrum antibiotics and prednisone was administered and resolved without sequelae. Three patients had grade 1 cutaneous toxicity: rash in two cases and pruritus in one. AEs present in one patient each were grade 2 oral candidiasis, and grade 1 asthenia, anemia, cough, flatulency, arthritis, and joint pain.

Five serious AEs were reported; four of them, namely, a femoral supracondylar fracture, a respiratory syncytial virus infection, an influenza A respiratory infection, and grade 2 persistent diarrhea, were considered unrelated to pembrolizumab and resolved without sequalae. The fifth one was the previously mentioned pneumonitis that was considered as possibly related to the study drug. There were no deaths attributable to pembrolizumab.

### 3.4. Biomarkers of Response

The initial hypothesis was that treatment responses to pembrolizumab could be determined by two factors: (i) the basal level of expression of PDL1 in residual pathological PC, and (ii) the characteristics of the patients’ immune system at enrollment and/or potential modulations induced by the treatment.

Given the specific inclusion criteria of the trial, 11 out of the 35 patients who signed the informed consent were deemed screening failures due to the lack of residual disease identifiable by multiparameter flow cytometry. We took advantage of this situation to investigate potential differences in the distribution of the different immune cell populations in BM and PB samples obtained at baseline from patients with and without detectable disease. Regarding BM, cases with persistent disease had a statistically significant lower percentage of CD8^+^ T lymphocytes (4.47% vs. 6.82%, *p* = 0.04), mainly at the expense of CD8^+^ effector T cells (1.78% vs. 2.71%, *p* = 0.01). Similar results were obtained in PB: CD8^+^: 9.27% vs. 18.92%, *p* = 0.02; CD8^+^ effector: 3.64% vs. 5.35%, *p* = 0.01, in cases with and without detectable residual disease, respectively (Figure 2).

Then, we sought to identify whether possible differences in patients’ basal immune profile could influence their behavior under treatment with pembrolizumab. No major differences were identified in the potentially responding patient either in the PD1/PDL1 axis basal expression or in the distribution of the different immune cell populations in BM and in PB as compared to the rest. Regarding the five early progressing patients, a statistically significant lower expression of PD1 in CD8^+^ T cells (mean fluorescence intensity MFI 755 vs. 995, *p* = 0.02) at the expense of CD8^+^ effector memory T cells (MFI 926 vs. 1327, *p* = 0.03) was observed when compared with the remaining cases (Figure 3A). Moreover, a similar pattern was observed in PB, with a lower expression of PD1 in CD8^+^ effector memory T cells, although differences did not reach statistical significance (MFI 548 vs. 761, *p* = 0.09) (Figure 3B).

We also aimed at investigating the specific changes induced by the pembrolizumab in PB samples (baseline vs. C3D1). Compared to baseline, treatment with pembrolizumab induced a decrease in the overall population of NK cells at C3D1 (3.39% vs. 2.29%, *p* = 0.03); specifically, we identified a statistically significant fall in the percentage of circulating and adaptive subsets after the treatment (0.61% vs. 0.43%, *p* = 0.007; 1.12% vs. 0.86%, *p* = 0.02) (Figure 4). However, no significant differences were present according to the ulterior patients’ response to pembrolizumab.

## 4. Discussion

In this study, we analyzed the efficacy and toxicity of pembrolizumab in patients with MM achieving a favorable response (≥VGPR) but with persistent residual disease after first- or second-line treatment. The rationale for this strategy was based on the results published by our group describing increased levels of PDL1 on persistent MRD-positive clonal PCs (as compared to normal PCs from healthy individuals) and a significant increase in PD1 expression among both CD4 and CD8 T cells [14]. From the 17 patients evaluable for efficacy, only one showed an upgrade in the response from VGPR to near CR (negative electrophoresis but persistent positive immunofixation and MRD-positive) that was maintained 30 months after entering the trial. In MM, there are no data available on the use of pembrolizumab as a single agent. In the study published by Lesokhin et al. [1] in which nivolumab was used in RRMM patients, only one of them responded to the treatment, while receiving concurrent local radiotherapy in one of the presenting plasmocytomas. In our trial, the responding patient upgraded his response at baseline, but residual disease persisted throughout the treatment period. It is important to highlight that disease stability was a prerequisite to enter the trial and that pembrolizumab was administered as a single agent, without steroids or another potentially active medication. However, the pattern of response (mainly happening after the first three cycles) and its limited quality (never reaching CR) does not allow ruling out a potential late therapeutic effect of ASCT. Patients included in this trial showed a median PFS of 14 months, which might seem shorter as compared to that observed in the GEM2012 trial [17]; however, patients enrolled in the latter study received consolidation (two cycles of bortezomib, lenalidomide, and dexamethasone) and maintenance (lenalidomide, dexamethasone ± ixazomib at least for 2 years). This added treatment and the fact that we also included patients treated after second line likely explain the commented differences in PFS between the two groups.

Although the number of patients in our study was limited, we were able to explore the specific toxicity of pembrolizumab as single agent in MM. In our trial, the dose and schedule of pembrolizumab were the same as in the KEYNOTE studies [3,4] and no steroids were used, which could have masked potential immune-related adverse reactions induced by the study drug. Among the AEs, some could be potentially immune-related, affecting usual organs such as the skin and gastrointestinal tract, as well as one episode of arthritis and one of pneumonitis, although none of them were in fact histologically confirmed as such. Two of them prompted treatment discontinuation: grade 3 diarrhea and grade 2 pneumonitis. Interestingly, all these AEs happened in patients assigned to the nonprogressing group, suggesting that, although this does not appear to deepen the quality of the response, their appearance may reflect a certain degree of immune-based activation associated with a better disease control [18,19,20].

Some other studies also assessed the safety and efficacy of checkpoint inhibition as consolidation following ASCT in patients with MM. One of them administered ipilimumab and nivolumab between 14 and 28 days post ASCT in patients with high-risk MM achieving at least stable disease after induction treatment. At 18 months post ASCT, authors reported a PFS of 71%, and 65% of patients developed immune-related AEs grade 2 or higher, where all of them but one resolved with systemic steroids (one patient died from pneumonitis related to study drugs) [21]. A second study enrolled 12 patients with high-risk MM that were treated with four cycles of pembrolizumab combined with lenalidomide plus or minus dexamethasone starting between 3 and 6 months post ASCT. With a median follow-up of 8.5 months when the study was reported, none of the patients had progressed, four patients (33%) had achieved sCR, and, of the four patients who completed therapy, two were MRD-negative by multiparametric flow cytometry. All patients had AEs of any grade, with 94% grade 1 or 2 and only one serious AE (*Haemophilus influenzae* pneumonia requiring inpatient admission) that was not thought to be related to pembrolizumab [22]. Both studies were halted after the FDA alert regarding the use of checkpoint inhibitors in MM.

An unexpected finding was the five patients that progressed in the first 4 months after enrollment. To better understand this behavior, we first confirmed that there were no statistically significant differences in the main clinical features between this group of patients (“early progressing”) and the rest that could account for their divergent clinical evolution. When we analyzed the immune profile of both groups, we found them completely superimposable, except for a lower PD1 expression in the T CD8^+^ effector memory subset in “early progressing” cases. Furthermore, although differences were not statistically significant, this unfavorable group of patients showed a lower expression of PD1 in almost all BM and PB immune cell populations, as well as a lower PDL1 in their BM clonal PCs as compared to the rest. This lower expression of the PD1/PL1 axis could have translated to a lower beneficial effect of pembrolizumab in this group of “early progressing” patients in whom the local immunosuppressive mechanisms involving PD1/PDL1 interactions would be less active as a mechanism of tumor resistance [23].

Two additional observations can be highlighted from the biological studies implemented in the trial. One is the significantly higher percentage of CD8^+^ T lymphocytes in cases without evidence of disease at baseline, both in BM and in PB, mainly at the expense of CD8^+^ effector T cells. This finding seems in line with the fact that the only definitive T cells shown to be protective in MM are clonal cytotoxic CD8^+^ T cells, which would be responsible for the disease control. Furthermore, this CD8^+^ T cell population has also been associated with a favorable prognosis and is invariably detected in long-term MM survivors [24,25,26]. Lastly, regarding the specific changes induced by pembrolizumab in PB at C3D1, no significant alterations were identified, except for a reduction in the percentage of NK cells according to the circulating and adaptive subsets. Little is known about the effects of immune checkpoint blockade on NK cells. In vitro, PD1 blockade enhanced cytotoxicity of expanded NK cells toward MM cells [27] but no studies regarding the influence of pembrolizumab on NK cells from patients with MM have been reported to date. In advanced non-small-cell lung cancer, during PD1 blockade, NKs progressively increased in patients achieving clinical benefit while they declined in nonresponding patients [28] In contrast, in melanoma, no consistent changes were identified in the proportion of NK cells after treatment with pembrolizumab [29,30].

In MM, the PD1/PDL1 blockade was not effective in the relapsed setting as monotherapy, and the subsequent phase III trials (KEYNOTE-185 and KEYNOTE-183) in combination with immunomodulatory agents had to be halted by the FDA because of an increase in deaths in the pembrolizumab arm. New combinations are currently being tried (i.e., with belantamab). We consider that PD1/PDL1 inhibition represents a specific mechanism of action that, if used with the adequate partners, with the correct population and timepoint, might have a role in the future management of MM. However, the current optimistic landscape of approved and other investigational agents in myeloma makes the field quite challenging for these types of agents.

## 5. Conclusions

In conclusion, pembrolizumab used as consolidation monotherapy was associated with an acceptable toxicity profile but did not upgrade the quality of the baseline responses in this MM patient population. Interestingly, our correlative immunophenotypic studies show that the group of patients progressing earlier under pembrolizumab consolidation were characterized by a lower PD1 expression in the T CD8^+^ effector memory subset.

## Figures and Tables

**Figure 1 cancers-12-03615-f001:**
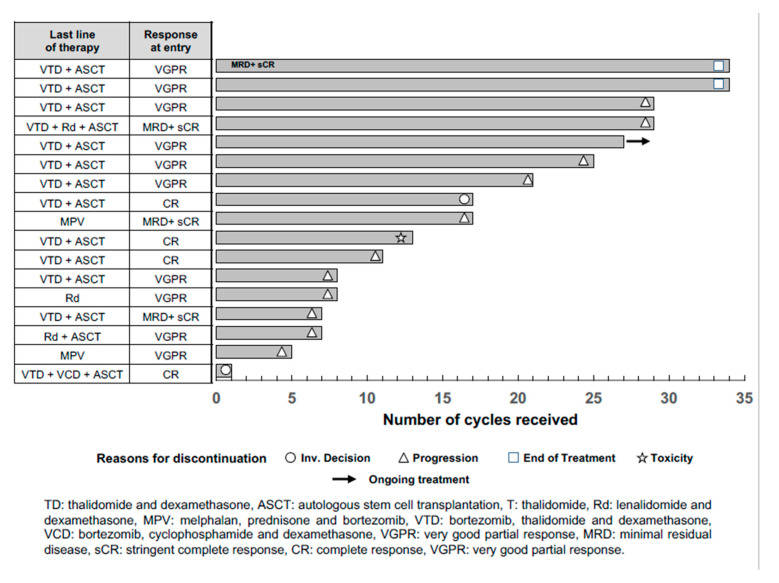
Clinical evolution of the 17 patients included in the study and evaluable for efficacy. Response at study entry, last line of therapy, and, when applicable, the prior line of treatment. Reasons for treatment discontinuation are depicted with specific symbols at the end of each patient’s representing bar.

**Figure 2 cancers-12-03615-f002:**
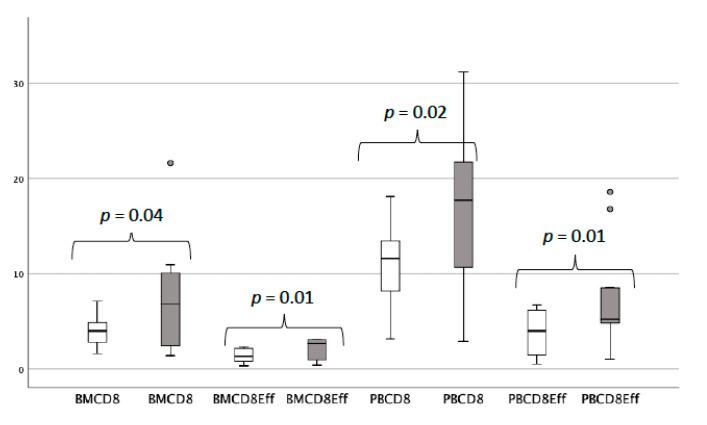
Percentages of the populations differently represented in patients with and without detectable disease by next generation flow at enrollment. White boxes represent the values obtained in minimal residual disease (MRD)-positive patients, whereas gray boxes correspond to MRD-negative patients.

**Figure 3 cancers-12-03615-f003:**
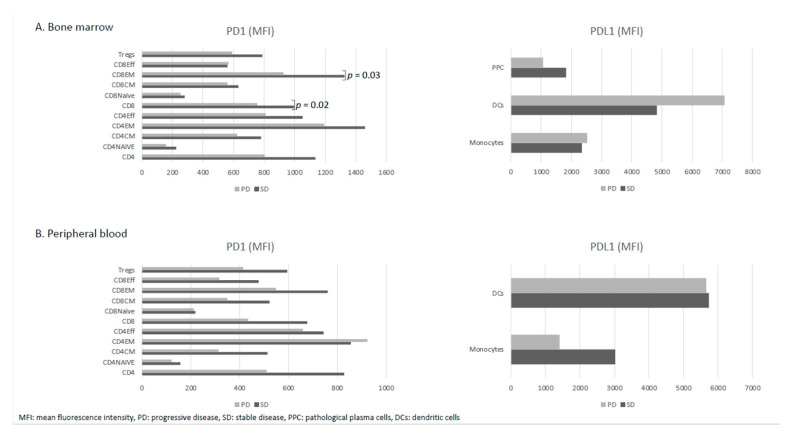
(**A**,**B**) Baseline PD1/PDL1 expression in the corresponding immune cell populations according to the patient’s behavior after treatment with pembrolizumab. PD1/PDL1 expression is represented as the mean fluorescence intensity (MFI) obtained in each cell population; “early progressing” patients are depicted with light-gray bars (PD) and “stable” patients are depicted with dark-gray bars (SD). Only statistically significant *p*-values are detailed.

**Figure 4 cancers-12-03615-f004:**
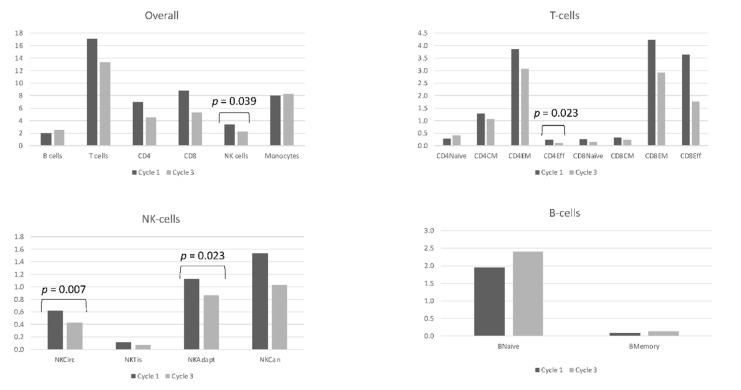
Percentages of the main immune cell populations and subpopulations in peripheral blood at baseline and after receiving two cycles of pembrolizumab. Dark-grey bars correspond to baseline values and light-gray bars correspond to those obtained at day 1 of cycle 3. Statistically significant differences in the percentage of cell populations between both timepoints are highlighted with their corresponding *p*-value.

**Table 1 cancers-12-03615-t001:** Patients’ characteristics.

Variable	Category	Treated		Efficacy-Evaluable	
		(*n* = 20)		(*n* = 17)	
Age, Years; Median (Range)		64 (43–78)		63 (44–78)	
Gender	Male	9	45%	7	41%
	Female	11	55%	10	59%
ECOG *	0	6	30%	5	29%
	1	13	65%	12	71%
MM isotype	IgG	9	45%	6	35%
	IgA	7	35%	7	41%
	BJ	4	20%	4	24%
ISS *	I	6	33%	6	40%
	II	6	33%	5	33%
	III	6	33%	4	27%
EMD		0	0%	0	0%
No. of prior lines	1	16	80%	13	77%
	2	4	20%	4	23%
Prior ASCT		16	80%	14	82%
Response at screening	VGPR	11	55%	8	47%
	CR	4	20%	4	24%
	sCR	5	25%	5	29%

MM: multiple myeloma, BJ: Bence-Jones, ISS: International Staging System, EMD: extramedullary disease, ASCT: autologous stem cell transplantation, VGPR: very good partial response, CR: complete response, sCR: stringent complete response; * one and two cases with missing information.

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
