# Peer review of "Pembrolizumab as Consolidation Strategy in Patients with Multiple Myeloma: Results of the GEM-Pembresid Clinical Trial"

_cancers, 2020, doi:10.3390/cancers12123615_

Round 1
Reviewer 1 Report
The authors have reported the results of a GEM-Permbresid clinical trial based on Permbrolizumab treatment in MM patients as a consolidation strategy in patients achieving at least VGPR but with persistent measurable disease. As is known that the addition of pembrolizumab to IMiDs based treatment (Pd or Rd) failed to improve clinical outcomes for patients with newly diagnosed or relapsed/refractory multiple myeloma, according to the results of KEYNOTE studies. FDA stopped those trials because of the serious adverse effects and treatment related death which alerted the future similar trials. Here, the trial itself was challenging and debatable. Researchers started this trial here based on their previous study which showed PD-L1/PD-1 presence in the tumor microenvironment and activity of PD-1 blockade in multiple myeloma mice model. They enrolled 17 patients with a very good partial response (VGPR) or better with measurable residual disease to receive pembrolizumab as consolidation. Their primary endpoint was the upgrade of the response obtained compared with the previous therapy. However, the results failed to their expectations. It seemed not a surprise for me. From my perspective, I would ask how this single agent consolidation affect in PFS and EFS in the long term.
Furthermore, the manuscript must be improved considerably to be understood by the readership of the journal.
- Complete list of abbreviations must be provided, and they must also be carefully defined at first time use. Line 115 (VGPR to ≥CR; CR to ≥ sCR; sCR flow MRD+ve to sCR flow MRD-ve) et al.
- Figures are obscure and unclear like in Figure 1 and 4.
- From the discussing part, did you find any related study in other PD-1/PD-L1 inhibitors?
- What’s your summary and outlook for the future study on PD-1/PD-L1 inhibitors in MM?
Reviewer 2 Report
Previous studies have shown no effect of check point inhibitors to treat myeloma and even some deleterious side effects when in combination with IMIDs and steroids. However, the Spanish co authors have found ex vivo in myeloma patients an increase of PD1 expression on CD4 T and CD8 T cells as well as PDL1 on oncogenic plasma cells when in a stage of positive MRD. Therefore, they conducted a clinical trial testing pembrolizumab for myeloma patients with minimal residual disease after one or two lines of treatment.
This is a multicenter national study evaluating the efficacy and safety of pembrolizumab monotherapy for patients with at least VGPR after 1 or 2 lines of therapy. The primary endpoint was upgrading the response. They did correlative immune studies by flow cytometry analyzing immune cells (T, NK, dendritic and plasma cells) from blood and bone marrow at different time points after treatment.
Twenty patients have been included and 20 analyzed for toxicity and 17 for response. Two patients stopped because of toxicity and only one patient out of 17 potentially improved his response following pembrolizumab treatment. Following the treatment, they observed a decrease in blood NK cells, especially the circulating and adaptive subsets.
Minor revisions
- The quality of figure 1 can be improved : it is difficult to read the attached table.
- Same for figure 3, too small. What is PPC ?
- Figure 2, add the p values
- Line 351 : write « non small cell lung cancer »
- Line 78 : cells
Round 2
Reviewer 1 Report
Dear Editors and authors,
Thanks for your response, I think you have improved the figure quality and data presentation, completed most of my questions.
I would like to recommend the paper for publication after detailed paper and figure organization. Take an example : 1. Line 182 It should be consistent with the rest paragraph. Indentation for the first line for all paragraph if you would. 2. Can you change the figure to a bigger texts for Figure 1, there is still a little bit obscure for me.
Best regards
Author Response
Thanks for your response, I think you have improved the figure quality and data presentation, completed most of my questions.
I would like to recommend the paper for publication after detailed paper and figure organization. Take an example :
1. Line 182 It should be consistent with the rest paragraph. Indentation for the first line for all paragraph if you would. Thank you very much for the recommendation and our apologies. We have indented the first line of all paragraphs and homogenized the structure of the paper.
2. Can you change the figure to a bigger texts for Figure 1, there is still a little bit obscure for me. Apologies once again for this. We have increased the size of the text in Figure 1. We will also send all the figures to the editorial to be adjusted in order to make them easily readable.